# Seasonal Study of Aflatoxin M1 Contamination in Cow Milk on the Retail Dairy Market in Gorgan, Iran

Hadi Rahimzadeh Barzoki [1], Hossein Faraji [2], Somayeh Beirami [1], Fatemeh Zahra Keramati [1], Gulzar Ahmad Nayik [3], Zahra Izadi Yazdanaabadi [4] and Amir Sasan Mozaffari Nejad [4,*]

[1] Department of Environmental Health Engineering, Faculty of Health and Environmental Health Research Center, Golestan University of Medical Sciences, Gorgan 49189-36316, Iran; rahimzadeh@goums.ac.ir (H.R.B.); sbeirami1359@gmail.com (S.B.)

[2] HSR Research Center, Vice-Chancellor for Health, Babol University of Medical Sciences, Babol 47176-47745, Iran; faraji_hoseyn@yahoo.com

[3] Department of Food Science and Technology, Government Degree College Shopian, Jammu and Kashmir 192303, India; gulzarnaik@gmail.com

[4] Universal Scientific Education and Research Network (USERN), JMU Office, Jiroft University of Medical Sciences, Jiroft 78616-15765, Iran; avaizadi2000@gmail.com

* Correspondence: asmnejad@jmu.ac.ir; Tel.: +98-34-43317802

**Abstract:** Milk and milk products are the main nutritional foods for all age groups, especially for infants and children. Milk may be dangerous to consume due to the presence of a harmful substance called Aflatoxin M1 (AFM1). The objective of this study was to assess the levels of AFM1 in milk, particularly those that exceed the standards set by the European Union (50 ng/L), the Food and Drug Administration (500 ng/L), and the Iranian National Standards Organization (100 ng/L). The study included one hundred and eighty samples of raw cow's milk from various retail dairy markets in Gorgan, with 45 samples collected during each season. The level of Aflatoxin M1 in the samples was measured using the enzyme-linked immunosorbent assay (ELISA) technique. AFM1 was detected in 139 (72.2%) raw cow milk samples with a range of 3.5–357 ng/L. All of the samples collected had Aflatoxin M1 concentration levels that were below the maximum limit of 500 ng/L set by the FDA. However, 41 samples (22.7%) exceeded the EU's limit of 50 ng/L, and 26 samples (14.4%) exceeded the INSO's limit of 100 ng/L for Aflatoxin M1 in raw cow's milk. The lowest and highest AFM1 levels of contamination were detected in the summer and winter seasons, which constituted 32 (71.1%) and 38 (84.4%) samples, respectively. The consumption of raw cow milk can lead to health risks for individuals from various age groups because regulatory limits are not being followed.

**Keywords:** Aflatoxin M1; ELISA; Gorgan; Iran; raw milk; seasons

## 1. Introduction

Milk is a fluid produced by female mammals and is widely recognized as a significant source of nutrition for humans. It is often considered one of the most complete foods, and its consumption is associated with several health benefits, including reducing blood pressure, preventing colon cancer and osteoporosis, and providing essential nutrients such as proteins, calcium, vitamins, and fatty acids [1–3]. Milk is not only consumed as a liquid, but it is also used in the preparation of various food items such as infant formulas, yogurt, cheese, chocolate, and pastries [1,4]. Although milk is a vital source of essential nutrients that support bodily functions, it is also a common source of humans' exposure to different contaminants. The collection and processing of milk can expose it to various contaminants, including pesticide residues, heavy metals, mycotoxins, hormones, and other substances that may enter the cow's system via feed or drug administration by producers. Consequently, milk may have hazardous materials of biological or chemical origin that can pose a risk to human health and food safety and security [1–6].

Mycotoxins are toxic secondary metabolites produced by species of filamentous fungi such as *Aspergillus* species, especially *A. parasiticus*, *A. nomius,* and *A. flavus* growing on seeds crops, and animal foraging during harvesting, storage, and processing [7–9]. Recent reports have shown that elevated climate temperatures and atmospheric $CO_2$ concentrations have been linked to increased mycotoxin production, as evidenced by a study on the occurrence of mycotoxins in agricultural products under varying weather conditions, which found that hot and dry weather was associated with a higher prevalence of mycotoxins [1,8,10].

Aflatoxins are a primary and poisonous class of mycotoxins, with over 300 varieties identified [4]. The major types of aflatoxins can be classified into four groups based on their fluorescence under blue or green light: aflatoxin B1 (AFB1), AFB2, AFG1, and AFG2. These toxins are predominantly found in agricultural cultures and food products, including cereal grains (such as maize, rice, pearl millet, wheat, barley, oats, and sorghum), spices (such as red pepper, black pepper, turmeric, cinnamon, ginger, and cumin), oilseeds (such as sunflower, groundnut, cottonseed, and soybean), and tree nuts (such as almonds, coconuts, peanuts, Brazil nuts, and pistachios) [8,9]. Additionally, AFM1 and AFM2 are other types and metabolic products that can be detected in the milk of lactating animals that have consumed feed contaminated with AFB1 and AFB2 [1,4,6].

AFM1 is generated in the liver through the activity of enzymes associated with microsomal Cytochromes P450 (CYP450). It is the monohydroxylated metabolite of AFB1. The presence of AFM1 in milk has been observed to have a direct relationship with the level of AFB1 in the animal's feed [11,12]. Studies investigating this connection have reported that approximately 0.3–6.2% of the ingested AFB1 in livestock is converted into aflatoxin M1 in milk. However, the rate at which AFB1 is transformed into AFM1 can vary among animals, from day to day, and even during different milking processes. Upon discontinuation of AFB1 intake, the concentration of aflatoxin M1 in milk decreases and becomes undetectable within 72 h [11–14]. Numerous studies have highlighted that milk and dairy products can potentially serve as sources of aflatoxin M1, posing a risk to human health, particularly for infants and children [15–17]. Therefore, aflatoxin B1 and aflatoxin M1 are both categorized as human carcinogens of Group 1 (Carcinogenic to humans) by the International Agency for Research on Cancer (IARC). Recently, AFM1 was reclassified from Group 2B (Possibly carcinogenic to humans) to Group 1 due to evidence that it can cause cancer [6].

The presence of AFM1 in milk products available on the market has prompted the need to implement measures for controlling AFM1 contamination, particularly in products intended for infants. The exposure of infants to AFM1 is a significant concern due to their high consumption of milk. Infants generally have a slower capacity for metabolizing toxins compared to adults, potentially leading to a prolonged presence of the toxin in their system and consequently affecting their growth during the neonatal stage. Clearly, infants are the population most vulnerable to the harmful effects of AFM1 [1,11,12,16].

Aflatoxins are subject to regulation in over 80 countries; however, there is a lack of international harmonization in their legislation [11]. The Iranian National Standards Organization (INSO) has set the maximum tolerable level of AFM1 in raw milk as 100 ng/L, while the European Union (EU) has set it at 50 ng/L [18,19]. The Food and Agriculture Organization (FAO)/World Health Organization (WHO) Joint Expert Committee on Food Additives (JECFA) and the Codex Alimentarius Commission also have a maximum limit of 50 ng/L for AFM1 in raw milk [20]. However, the Food and Drug Administration (FDA) has set a higher limit of 500 ng/L for AFM1 in raw milk [21]. The legal limits for aflatoxin M1 can vary among different authorities due to factors such as geography, agricultural practices, and climate, and these limits are subject to change and may differ by country or region [6,22].

Detecting and quantifying aflatoxin M1 is of utmost importance due to its toxicity and the established maximum residue levels. Researchers have shown great interest in developing accurate and efficient methods for mycotoxin analysis. Various analytical techniques have been developed, each offering different levels of sensitivity and accuracy, catering to

specific purposes and requirements [1,11,12]. AFM1 analysis is currently conducted using various methods, including thin-layer chromatography (TLC), high-performance liquid chromatography (HPLC), and enzyme-linked immunoassays (ELISA). However, ELISA, despite its simplicity, has some drawbacks, such as lengthy incubation periods and multiple washing and mixing steps. Consequently, researchers have developed modified ELISA methods in recent years to enhance the detection of AFM1 in milk and dairy products. In Iran, the ELISA method is widely employed by researchers due to its user-friendly nature, rapidity, and automation potential [1,4,6,7,12].

The purpose of the present study was to (1) analyze the levels of AFM1 in raw milk using ELISA, (2) compare these levels to the maximum limits set by the EU, the FDA, and the Iranian Standard (INSO), (3) evaluate the seasonal occurrence of this mycotoxin in raw milk in Gorgan, Iran, and (4) bring the issue to the attention of public health authorities for the extensive monitoring and regulation of mycotoxins. The study results will be useful for farmers, merchants, and consumers in the community.

## 2. Materials and Methods

### 2.1. Sampling

A total of 180 raw cow milk samples were obtained randomly from bulk tanks of milk from the different retail dairy product markets of Gorgan in the Golestan province of Iran. All of the samples were at least 1 L each and were collected during October 2022 (autumn indicator), December 2022 (winter indicator), March 2023 (spring indicator), and June 2023 (summer indicator). All samples were transported to the laboratory in an icebox at 4 °C, and then the samples were frozen at −20 °C until analysis.

### 2.2. AFM1 Analysis by ELISA

An AFM1 competitive ELISA kit (Ridascreen AFM1 Art. No.: R1121, R-Biopharm, Darmstadt, Germany) was used to quantitatively analyze AFM1 in the raw milk samples. Sample preparation was followed according to the instructions suggested by the ELISA kit (R-Biopharm, Darmstadt, Germany).

### 2.3. Sample Preparation

In the first step, the raw milk samples were thawed at room temperature (20–25 °C) and subsequently centrifuged at 10 °C and $3500 \times g$ for 10 min. Following centrifugation, the upper fat layer was removed by aspiration with a Pasteur pipette, and 100 µL of the remaining sample was directly used per well for the AFM1 analysis.

### 2.4. ELISA Test Procedure

The kits were used according to the manufacturer's instructions: AFM1-antibody-coated microtiter plate (supplied with the kit) was pipetted into each well (100 µL/well) and incubated at room temperature (20–25 °C) for 15 min. The liquid was poured out of the wells and the microwell holder was tapped upside down vigorously (three times in a row) against absorbent paper to ensure the complete removal of liquid from the wells. All the wells were filled with 250 µL of wash buffer, and then, the liquid was poured out, again. The washing procedure was reported twice. Then, 100 µL of a standard solution and the prepared samples were added to separate wells and incubated for 30 min at room temperature in the dark. The microplate wells were washed twice with approximately 250 µL of wash buffer per well. Next, 100 µL of the enzyme conjugate was added to each well of the used plate and incubated for 15 min at room temperature in the dark. Afterward, the microplate wells were wash twice with approximately 250 µL of wash buffer per well. A 100 µL of substrate solution was added into the wells, and the reaction was allowed to proceed in the dark for 15 min at room temperature. Following the addition of 100 µL of the stop solution to each well, the absorbance was measured photometrically at 450 nm against an air blank and by using an enzyme-linked immunosorbent assay reading apparatus. All tests were performed in duplicate. The calibration curve was virtually linear

in the 5–100 ng/L range (Figure 1). The AFM1 concentration in ng/L corresponding to the extinction of each sample was read from the calibration curve. The detection limit of the analysis was 5 ng/L. Recoveries were determined in milk samples spiked at levels of 5–100 ng/L. The mean recovery and coefficient of variation were 90 and 10%, respectively.

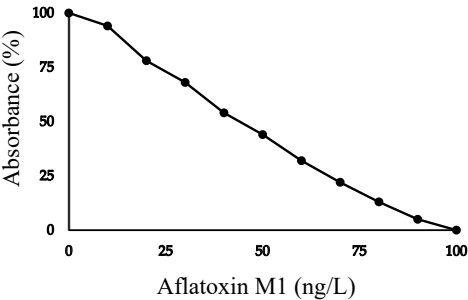

**Figure 1.** Calibration curve of AFM1.

*2.5. Statistical Analysis*

SPSS software version 19 (ISPSS, Chicago, IL, USA) was used for the results of the statistical analysis. The mean level of AFM1 in the raw milk samples collected during the different seasons was compared using the ANOVA test. The mean concentration of AFM1 in the samples and the permitted amount of this mycotoxin according to the INSO and EU regulations are 100 and 50 ng/L, respectively. A significance level of $p < 0.05$ was used.

**3. Results**

Table 1 summarizes the number of raw milk samples analyzed and the number of samples found to contain detectable levels of AFM1 contamination from Gorgan, Iran. Of 180 samples, 139 (77.2%) contained AFM1 in concentrations ranging from 3.5 to 357 ng/L. As shown in Table 1, none of the raw milk samples had AFM1 concentrations above the highest tolerance limit (500 ng/L) set by the FDA. However, 41 samples (29.5% of the positive samples) contained concentrations above 50 ng/L which is the tolerance limit adopted by the EU and the Codex Alimentarius Commission for liquid milk. In addition, the AFM1 concentration in 26 samples (18.7% of the positive samples) was higher than the maximum limit of 100 ng/L set by the INSO regulations.

**Table 1.** Occurrence of aflatoxin M1 in raw cow milk in Gorgan, Iran.

| Season | Sample Tested *n* | Distribution of Samples *n* (ng/L) | | | | | | Exceed Legal Limit *n* (%) | | |
|---|---|---|---|---|---|---|---|---|---|---|
| | | <5 | 5–50 | 51–150 | 151–250 | 251–500 | >500 | INSO [1] | EU [2] | FDA [3] |
| Spring | 45 | 3 | 19 | 8 | 3 | 1 | 0 | 8 | 12 | 0 |
| Summer | 45 | 2 | 24 | 3 | 3 | 0 | 0 | 4 | 6 | 0 |
| Autumn | 45 | 2 | 27 | 4 | 2 | 0 | 0 | 4 | 6 | 0 |
| Winter | 45 | 0 | 21 | 10 | 4 | 3 | 0 | 10 | 17 | 0 |
| **Total (*N*)** | **180** | **7** | **91** | **25** | **12** | **4** | **0** | **26 (14.4)** | **41 (22.7)** | **0** |

[1] The Iran National Standards Organization limit for AFM1 in milk is 100 ng/L. [2] The European Union limit for AFM1 in milk is 50 ng/L. [3] The Food and Drug Administration limit for AFM1 in milk is 500 ng/L.

The distribution by season of cow raw milk samples and AFM1 contamination is presented in Table 2. The concentrations of AFM1 varied among the seasons, with them being 3.5–237 ng/L 5.4–357 ng/L, 4.1–243 ng/L, and 4.5–241 ng/L for the autumn, winter, spring, and summer samples, respectively with mean values of 36.82 ± 57.54 ng/L, 78.83 ± 93.24 ng/L, 63.02 ± 72.64 ng/L and 43.38 ± 60.55 ng/L, respectively. At a significance level of $p < 0.05$, there is a notable variation in the mean levels of AFM1 concentrations across different seasons. The mean levels of AFM1 during the summer and autumn are noticeably less than the average levels of AFM1 during the winter and spring.

**Table 2.** Distribution by season of cow raw milk samples and AFM1 concentration (ng/L) in Gorgan, Iran.

| Season | No. of Samples | | Aflatoxin M1 Concentration (ng/L) | | |
|---|---|---|---|---|---|
| | Tested *n* | Positive *n* (%) | Minimum | Maximum | Mean $\pm$ SE |
| Spring | 45 | 34 (75.5) | 4.1 | 243 | 63.02 $\pm$ 72.64 |
| Summer | 45 | 32 (71.1) | 4.5 | 241 | 43.38 $\pm$ 60.55 |
| Autumn | 45 | 35 (77.7) | 3.5 | 237 | 36.82 $\pm$ 57.54 |
| Winter | 45 | 38 (84.4) | 5.4 | 357 | 78.83 $\pm$ 93.24 |
| **Total** (*N*) | **180** | **139 (77.2)** | **3.5** | **357** | **56.32 $\pm$ 74.37** |

## 4. Discussion

Milk and dairy products provide essential nutrients and are consumed globally across various age groups. They are particularly important for infants, children, and the elderly [1,22]. These products have been studied extensively and are recognized as valuable contributors to a healthy diet. However, the presence of AFM1, a known carcinogen, in milk and dairy products poses significant health risks, including liver cancer. Despite research efforts, many countries, including Algeria, Bangladesh, Brazil, China, Croatia, Greece, Iran, Morocco, Pakistan, Sudan, Turkey, and some African countries, continue to report high levels of AFM1 contamination. Several studies in Iran have examined AFM1 levels in different dairy products, revealing instances of levels exceeding permissible limits [4,6,22–31].

As shown in Table 2, aflatoxin levels were reported from highest to lowest in the winter, spring, summer, and autumn, respectively. Additionally, the statistical analysis of the data showed that AFM1 was significantly higher in the winter than in the autumn and summer ($p \leq 0.05$). This could be associated with the quantity of mixed feed ingested in each season. The reason that there is a higher amount of toxins in the winter season is that there is not enough fresh forage for animal feed in the winter, and farmers should use feed concentrate and stored feed [26]. Therefore, environmental conditions such as temperature and humidity may be more suitable in this season for contamination with toxigenic *Aspergillus* fungi such as *A. flavus* and *A. parasiticus* and the formation of aflatoxins [32]. Consistent with our results, a previous study by Mahosotanand [33] screened the AFB1 contamination of mixed feed collected from different dairy farms and found that the highest AFB1 concentration was detected in the winter compared with the rainy season and summer. The amount of aflatoxin was higher in the winter than in the spring, although this difference was not statistically confirmed ($p > 0.05$), because the justification is that fodder is planted and grown in the spring, and in the early spring they still use the fodder stored in the winter, and the fresh fodder that is obtained in the spring is mostly at the end of the spring and reaches the animal feed in the summer [34]. Our findings are consistent with the recent research conducted by Bervis et al. [35] and Mohammedi-Ameur et al. [36]. This could possibly be attributed to the exceptionally hot summer, severe drought, warm autumn, and insufficient rainfall in the winter season of the year. This study revealed that the highest incidence of aflatoxin M1 in the raw milk samples was detected in the winter season. However, this was similar to several previous studies from several regions of Iran and also other countries [3,13,25,37–42].

In this survey, 139 (77.2%) of the 180 raw cow milk samples were contaminated with AFM1 and their mean toxin concentration was 56.32 $\pm$ 74.37 (ng/L); however, the AFM1 levels in 22.7% of the samples were higher than the tolerance level regulated by the EU. Of all the raw milk samples, only 26 (14.4%) had AFM1 levels higher than the INSO.

Our results are comparable with some previous investigations carried out in Iran and other countries. Table 3 shows the compilation of data for the detection of AFM1 from previous studies in Iran and several countries that were measured by enzyme-linked immunosorbent assay (ELISA), thin layer chromatography (TLC), and high-performance liquid chromatography (HPLC) methods.

**Table 3.** Occurrence and levels of aflatoxin M1 in raw cow milk samples published in previous studies.

| Location | No. of Samples | Positive *n* (%) [1] | Detection Method | Range (ng/L) | Exceed Legal Limit *n* (%) | | Reference |
|---|---|---|---|---|---|---|---|
| | | | | | EU [2] | FDA [3] | |
| Iran | 60 | 44 (73) | ELISA | 3.6–41.95 | 18 (30) | 0 | Kamkar et al. [13] |
| Iran | 254 | 204 (80.3) | ELISA | 11–321 | 144 (56.7) | NR [4] | Fallah et al. [39] |
| Serbia | 678 | 678 (100) | ELISA | 25–1000 | 382 (56.3) | 167 (24.6) | Tomasevic et al. [43] |
| Iran | 64 | 54 (84.3) | ELISA | 61–188.2 | 23 (35.9) | NR | Bahrami et al. [38] |
| Italy | 416 | 51 (12.3) | ELISA | 4–52 | 0 | NR | De Roma et al. [44] |
| Kenya | 96 | 96 (100) | ELISA | 15.4–4563 | 64 (66.6) | 7 (7.3) | Kuboka et al. [26] |
| Iran | 257 | 123 (47.9) | HPLC | LOD > 150 | 4 (1.6) | 0 | Khaneghahi Abyaneh et al. [45] |
| Egypt | 60 | 13 (21.6) | TLC | 50–660 | 13 (21.6) | 3 (5) | Ismaiel et al. [46] |
| Cyprus | 1026 | NR | HPLC | 0–96 | 117 (11.4) | NR | Tuncay and Oniz [42] |
| Spain | 60 | 23 (38.3) | HPLC | 8–67.2 | 5 (8.3) | NR | Bervis et al. [35] |
| China | 195 | 128 (66.7) | ELISA | 5–191 | 6 (3.1) | NR | Zheng et al. [47] |
| Ghana | 120 | 67 (55.8) | HPLC | 0–3520 | 63 (52.5) | 50 (41.7) | Kortei et al. [48] |
| Bangladesh | 90 | 39 (43) | ELISA | 23.08–533.83 | 27 (30) | 2 (5) | Ali et al. [23] |
| Hungary | 278 | 191 (68.7) | ELISA | 5–173 | 26 (9.4) | NR | Buzas et al. [25] |

[1] Percent (%): All percentages reported are based on the entire sample set. [2] The European Union limit for AFM1 in milk is 50 ng/L. [3] The Food and Drug Administration limit for AFM1 in milk is 500 ng/L. [4] NR: not Reported.

In a related study from Iran, all 48 raw cow milk samples were contaminated with AFM1 (mean: 27.08 ± 3.95 ng/L). Furthermore, 10 (20.83%) samples contained greater than the maximum limit set by the EU, but no samples were found exceeding the INSO limit [49]. Lower values were reported by Ismaiel et al. [46] in raw milk from Egypt, who found that 11 (18.3%) out of 60 raw milk samples analyzed by the TLC method were contaminated and 10 (90.9% of the positive samples) exceeded the limit established by the EU; additionally, 1 (9.1% of the positive samples) had a concentration above the FDA limit. However, this study was in contrast to our findings which showed that none of the raw milk samples had AFM1 concentrations above the highest tolerance limit set by the FDA. In a previous study carried out in Turkey, Ozturk Yilmaz and Altinci [29] reported that in 16 (61.5%) out of 26 milk samples, AFM1 was detected in concentrations between 3 ng/L and 47.81 ng/L by the HPLC method and that none of the samples exceeded the EU and Turkish regulations (50 ng/L), while in the current study, we reported that the AFM1 content in 41 samples of raw cow's milk was above the EU limit. A study by Patyal et al. [50] from India using the ELISA method revealed that 70 out of 116 raw milk samples (60%) were contaminated with AFM1. Additionally, they found that 53% of these samples exceeded the maximum limit established by the EU, while 47% were above the regulatory limit set by the FDA. In contrast, our results show that none of the detected samples exceeded the limit set by the FDA.

According to a study reported by Jedidi et al. [41], of 20 raw cow's milk samples, collected in Algeria, 4.7% (1/20) were positive for AFM1 with a value of 5.8 ng/L detected by the ELISA method. Furthermore, no samples were above the permissible level according to the EU. However, the result contrasts with our findings which showed that 77.2% (139/180) of samples were contaminated with AFM1 and that 41 samples also exceeded the EU limit. A study from Iran by Ahmadi [37] using the ELISA method showed that 37 (82.2%) out of 45 raw milk samples were contaminated with AFM1. Additionally, 18 (40%) samples had higher aflatoxin M1 content than the limit allowed by the FDA, but our results showed that no sample examined exceeded the limit set by the FDA. Another similar study from Iran by Ghajarbeygi et al. [40] revealed that 34/60 (56.6%) raw cow milk samples contained aflatoxin M1 with a range of concentration from 62.5 to 127.87 ng/L using the ELISA method. In addition, the AFM1 concentration in 18 (30%) and 3 (5%) raw milk samples was higher than the maximum limit of 50 ng/L and 100 ng/L set by the EU and the Iranian regulations limit, respectively. Moreover, no samples were found above the FDA legal limit regulations for AFM1 concentrations in milk, which was similar to our current results.

In a previous study conducted in Brazil using the HPLC technique to determine AFM1, Corassin et al. [51] reported that 35 out of 40 (87.5%) raw milk samples were contaminated with AFM1 and that 4 (10%) of those had concentrations above the EU's recommended limit, but in this current study, 139 (77.2%) samples of raw milk were contaminated with AFM1, and 41 (22.7%) of all samples were observed to have AFM1 concentrations above the EU limit. In a similar survey conducted by Zebib et al. [52], it was found that all 64 raw milk samples from Ethiopia tested positive for AFM1 using the ELISA method. Additionally, they found that 62.50% of the samples exceeded the maximum limit established by the EU and that 21.87% were above the regulatory limit set by the FDA. Another similar study from Pakistan revealed that 18/28 (64.2%) raw cow milk samples contained aflatoxin M1 with the highest mean of 82.4 ± 7.8 ng/L using the HPLC method. Moreover, 7 (25%) of the samples had detected levels of AFM1 higher than the permissible limit of the EU. In addition, no samples were found to be above the FDA regulations for AFM1 concentrations in milk which was similar to our current results [53].

## 5. Conclusions

The present research reports that a high percentage of raw cow milk samples collected in Gorgan in Golestan Province in the northeast of Iran and the southeast of the Caspian Sea contains AFM1, with a prevalence of 77.2%. Of these samples, 14.4% and 22.7% were

found to exceed the regulatory limits set by the INSO and EU, respectively. The findings of this study have significant implications for farmers and scientists in other countries who are concerned about AFM1 contamination in dairy products. The importance of ensuring high-quality animal feed, involving all stakeholders in the value chain such as farmers, cooperatives, feed traders, regulatory agencies, and public health organizations cannot be overstated. By implementing strict regulations and monitoring measures at every stage of the feed and milk value chain, it is possible to effectively address the issue of AFM1 contamination. These findings highlight the necessity of regular inspections of milk and milk products, as well as rigorous regulation of livestock feed, to mitigate the occurrence of AFM1 contamination and ensure safe dairy products. Therefore, these research outcomes can serve as a valuable reference and guide for farmers and scientists worldwide to adopt best practices and preventive measures.

**Author Contributions:** Conceptualization, H.R.B. and A.S.M.N.; methodology, H.F., S.B. and F.Z.K.; software, H.R.B., G.A.N. and A.S.M.N.; validation, H.R.B., H.F. and S.B.; formal analysis, H.R.B. and A.S.M.N.; investigation, S.B., F.Z.K. and Z.I.Y.; resources, H.R.B., S.B. and H.F.; data curation, H.R.B. and A.S.M.N.; writing—original draft preparation, G.A.N., Z.I.Y. and A.S.M.N.; writing—review and editing, G.A.N. and A.S.M.N.; visualization, H.R.B. and A.S.M.N.; supervision, H.R.B. and A.S.M.N.; project administration, H.R.B. and A.S.M.N.; funding acquisition, H.R.B. All authors have read and agreed to the published version of the manuscript.

**Funding:** This research was funded by Golestan University of Medical Sciences, grant number IR.GOUMS.REC.1401.268. The APC was waived by the journal.

**Institutional Review Board Statement:** Not applicable.

**Informed Consent Statement:** Not applicable.

**Data Availability Statement:** The data will be made available when requested.

**Acknowledgments:** The authors greatly thank the Vice-Chancellor of Research and Technology of Golestan University of Medical Sciences and Health Services for financial supporting this project.

**Conflicts of Interest:** The authors declare no conflict of interest.

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
