# Peer review of "Seasonal Study of Aflatoxin M1 Contamination in Cow Milk on the Retail Dairy Market in Gorgan, Iran"

_2624-862X, doi:10.3390/dairy4040039_

Round 1

Reviewer 1 Report

The paper presents a study on the occurrence and risk assessment of aflatoxin M1 (AFM1) in raw cow milk samples collected from Gorgan, Iran. The authors used ELISA method for AFM1 analysis and compared the levels with regulatory limits set by INSO, EU, and US FDA. The study found a high prevalence of AFM1 contamination in raw cow milk samples, with a significant percentage exceeding the regulatory limits. The paper emphasizes the need for strict regulations and monitoring measures to reduce AFM1 contamination in dairy products.

Overall Evaluation: The paper provides valuable insights into the occurrence and risk assessment of AFM1 in raw cow milk in Gorgan, Iran. The study methodology is well-described, and the results are presented clearly. The discussion and conclusion sections effectively highlight the implications of the findings and suggest measures to address AFM1 contamination. However, further details on the sampling strategy and limitations of the study could enhance the paper's overall quality. Moreover, authors must describe how they ensure that ELISA method were calibrated. The validation method was properly carried out?

Author Response

Dear Reviewer:

Greetings,

Thank you for your comment and your time. We sincerely appreciate your valuable feedback. We have addressed the sampling strategy by including it in Lines 124-125. Additionally, we have incorporated the validation method in Lines 153-159, both of which are highlighted in yellow with red font.

In the current study, we acknowledge that there was a limitation concerning the analysis of the samples. Due to budget constraints, we faced challenges in proceeding with the HPLC analysis. Moreover, considering the large number of samples, we made the decision to utilize the ELISA method for analysis. We understand that the use of the ELISA method may have impacted the evaluation, and we acknowledge this limitation in our study. Despite these limitations, we firmly believe that the results obtained through the ELISA method provide valuable insights, as highlighted in yellow with red font in Lines 153-159.

Once again, we extend our gratitude for your constructive feedback and suggestions.

Faithfully Yours,

AS Mozaffari Nejad (Ph.D.)

On behalf of the other Authors

Reviewer 2 Report

In the Introduction, it would be more appropriate to give citations for specific information (sentences) rather than for entire paragraphs so that it is possible to find which data belongs to which source. It applies primarily to particular information, not to generally valid information. For example, in Lines 69-75, the reader might be interested in which of the five articles contains specific information about the decrease of AFM1 in 72 hours. I do not assume that such specific information is given in all five mentioned articles. So, I recommend authors to be more specific in their references.

Lines 66-80 – Use abbreviations for all aflatoxins mentioned above.

Lines 79-80 - I recommend adding information about what Groups 1 and 2B mean for clarification, like this: Group 1 (proven to be carcinogenic to people), Group 2B (probably carcinogenic to people).

Lines 92-93 – full name is the Joint FAO/WHO Expert Committee on Food Additives. The authors can also add an abbreviation – JECFA.

Line 94 – An abbreviation (FDA or US FDA) should be added in this place. Only one form of abbreviation must be used.  The authors inappropriately alternate two abbreviations of this institution - FDA and US FDA.

Lines 99, 125, 126, 154, 156, 171, 210, 212 and so on – use AFM1 instead of full name. Check the use of abbreviations in the whole text.

Line 170 – Use INSO; do not repeat full name and abbreviation.

Lines 182-184 - The explanations under Table 1 - US FDA and INSO abbreviations are unnecessary, especially when the EU abbreviation is not given. Note 3 - Use singular instead of plural (limits for AFM1 - correctly limit for AFM1), similarly, in Table 3.

Lines 190-207 – I think this part is too long. Some sentences belong to the Introduction rather than to the discussion part. Please revise this long paragraph to clarify what you want to say. Lines 206-207 – What did authors want to express with this sentence - However, the lowest articles are available in the northeast of Iran and southeast of the Caspian Sea. A reference source is missing in this unclear information.

Lines 211-212 – However, our result was similar to several previous studies that reported higher concentrations of aflatoxin M1 in cool seasons than in warm seasons. The authors should add in which studies it was stated. Without citing sources, the information is vague.

Line 219 – use AFB1.

Line 232 – cow instead caw

Line 247 – add units - AFM1 (mean: 27.08 ± 3.95).

I think repeating the limits (EU – 50 ng/L; INSA – 100 ng/L; US FDA – 500 ng/L) too often is somewhat unnecessary. Moreover, the distinction between the EU limit as the strictest, the INSA limit as the medium, and the US FDA as the most benevolent of the limits is relatively easy to remember. However, this is only my opinion, and the authors may disagree.

Some sentences in the discussion are cumbersome, which can lead to difficulty in understanding the text and the information it conveys. Therefore, the text in the Discussion needs to be revised and its comprehensibility improved. For example - In a similar survey by Zebib et al [47] found 100% positive samples of 64 raw milk samples were contaminated with AFM1 by the ELISA method from Ethiopia. It is incomprehensible.

New data on the presence of contaminants in food are needed. In this regard, the article provides an extension of information on the content of aflatoxin M1 in raw milk from a specific region of Iran. I appreciate the summary in Table 3, which conveniently presents several studies and their comparison with the permissible limits. On the other hand, the discussion part could be better written, and its language revision would help the readability and understanding of the entire text.

Author Response

Reviewer 2

Dear Esteemed Reviewer,

Greetings,

We would like to express my sincere gratitude for dedicating your valuable time to review our manuscript titled "Seasonal study of Aflatoxin M1 contamination in cow milk of retail dairy market in Gorgan, Iran." Your insightful feedback and profound insights have played a pivotal role in enhancing the quality of our research. We deeply appreciate your recognition of our work.

Your feedback has been constructive, and we have taken it to heart. We have carefully considered your suggestions and made the necessary revisions to ensure a more comprehensive documentation of our manuscript. Your expertise and attention to detail have been invaluable in refining our work.

Once again, I want to express my heartfelt appreciation for investing your time, effort, and expertise in the review of our manuscript. Your contribution has undoubtedly contributed to the advancement of our research.

Thank you sincerely.

Faithfully yours,

Authors

Reviewer Comments:

In the Introduction, it would be more appropriate to give citations for specific information (sentences) rather than for entire paragraphs so that it is possible to find which data belongs to which source. It applies primarily to particular information, not to generally valid information. For example, in Lines 69-75, the reader might be interested in which of the five articles contains specific information about the decrease of AFM1 in 72 hours. I do not assume that such specific information is given in all five mentioned articles. So, I recommend authors to be more specific in their references.

  • Thank you very much for the useful comments. We have made changes to address your concern by highlighting the revisions in red font and yellow highlighting in the text.

Lines 66-80 – Use abbreviations for all aflatoxins mentioned above.

  •  

Lines 79-80 - I recommend adding information about what Groups 1 and 2B mean for clarification, like this: Group 1 (proven to be carcinogenic to people), Group 2B (probably carcinogenic to people).

  • Thank you for your attention! We have made the requested additions at Line 77 and Line 79.

Lines 92-93 – full name is the Joint FAO/WHO Expert Committee on Food Additives. The authors can also add an abbreviation – JECFA.

  • Thank you. Revised and shown at Lines 91-93.

Line 94 – An abbreviation (FDA or US FDA) should be added in this place. Only one form of abbreviation must be used.  The authors inappropriately alternate two abbreviations of this institution - FDA and US FDA.

  • The comments you provided were greatly appreciated. Thank you! We have carefully reviewed your suggestions and made the necessary changes throughout the entire text, replacing the previous term with 'FDA' as you recommended.

Lines 99, 125, 126, 154, 156, 171, 210, 212 and so on – use AFM1 instead of full name. Check the use of abbreviations in the whole text.

  • We have revised the text as per your suggestions, and the changes are indicated with red font and yellow highlighting.

Line 170 – Use INSO; do not repeat full name and abbreviation.

  •  

Lines 182-184 - The explanations under Table 1 - US FDA and INSO abbreviations are unnecessary, especially when the EU abbreviation is not given.

  • Revised according to your suggestion. Thank U.

Note 3 - Use singular instead of plural (limits for AFM1 - correctly limit for AFM1), similarly, in Table 3.

  •  

Lines 190-207 – I think this part is too long. Some sentences belong to the Introduction rather than to the discussion part. Please revise this long paragraph to clarify what you want to say.

  • Corrected and shown at Lines 197-205.

Lines 206-207 – What did authors want to express with this sentence - However, the lowest articles are available in the northeast of Iran and southeast of the Caspian Sea. A reference source is missing in this unclear information.

  • Thank you for your comments. We deleted.

Lines 211-212 – However, our result was similar to several previous studies that reported higher concentrations of aflatoxin M1 in cool seasons than in warm seasons. The authors should add in which studies it was stated. Without citing sources, the information is vague.

  • We have deleted the mentioned information because it was previously reported at Lines 228-229.

Line 219 – use AFB1.

  •  

Line 232 – cow instead caw

  • Thank you. Corrected.

Line 247 – add units - AFM1 (mean: 27.08 ± 3.95).

  • We added the Unit ng/L at Line 246.

I think repeating the limits (EU – 50 ng/L; INSA – 100 ng/L; US FDA – 500 ng/L) too often is somewhat unnecessary. Moreover, the distinction between the EU limit as the strictest, the INSA limit as the medium, and the US FDA as the most benevolent of the limits is relatively easy to remember. However, this is only my opinion, and the authors may disagree.

  • Thank you so much for your attention about them. We deleted some of them according to your comments.

Some sentences in the discussion are cumbersome, which can lead to difficulty in understanding the text and the information it conveys. Therefore, the text in the Discussion needs to be revised and its comprehensibility improved. For example - In a similar survey by Zebib et al [47] found 100% positive samples of 64 raw milk samples were contaminated with AFM1 by the ELISA method from Ethiopia. It is incomprehensible.

  • We have reviewed the text and deleted some sentences that were difficult to understand. Additionally, we made changes to Lines 283-284 and highlighted them accordingly.

New data on the presence of contaminants in food are needed. In this regard, the article provides an extension of information on the content of aflatoxin M1 in raw milk from a specific region of Iran. I appreciate the summary in Table 3, which conveniently presents several studies and their comparison with the permissible limits. On the other hand, the discussion part could be better written, and its language revision would help the readability and understanding of the entire text.

  • Thank you for your comments and your encouragement regarding Table 3. We genuinely appreciate your feedback. We understand your suggestion about the discussion part and the need for language revision to improve the overall readability and understanding of the text. We have carefully reviewed and revised the discussion section to ensure clarity and coherence. Additionally, we will take your comments into consideration for future studies to further enhance the quality of our research. Your valuable input will undoubtedly be useful for other researchers in the field. Once again, thank you for your constructive feedback.

Reviewer 3 Report

REVIEW for the journal Dairy (ISSN 2624-862X)

Article “Seasonal study of Aflatoxin M1 contamination in cow milk of retail dairy market in Gorgan, Iran

Manuscript ID: dairy-2555566

Authors:  Hadi Rahimzadeh Barzoki, Hossein Faraji, Somayeh Beirami, Fatemeh Zahra Keramati, Gulzar Ahmad Nayik, Zahra Izadi Yazdanaabadi, Amir Sasan Mozaffari Nejad 

Brief summary. Dairy products, particularly milk, are essential components of nutrition. However, it's crucial to be aware that milk can potentially pose health risks due to the presence of a harmful substance known as Aflatoxin M1 (AFM1). This study aimed to evaluate the AFM1 levels in cow's milk available in the retail dairy market of Gorgan, Iran. The assessment was conducted in accordance with the guidelines established by the European Union, the U.S. Food and Drug Administration, and the Iranian National Standards Organization (INSO). The accomplishment of these objectives holds practical relevance, particularly concerning food safety in the mentioned region and its implications for human health.

General concept comments

1.       The introduction provides an overview of literature sources related to the examined topic (22 articles), outlines the article's objectives, with an emphasis on highlighting the practical significance of the conducted research.

·         However, I noticed that there is a lack of a clear hypothesis formulation and the authors' perspective on how their article could be relevant in theoretical terms to the international scientific community?

2.       Materials and Methods. The "Materials and Methods" section encompassed several subsections: Sampling, AFM1 analysis by ELISA, Sample preparation, ELISA test procedure, and Statistical analysis. The sample consisted of 180 raw cow milk samples. Statistical analysis employed the ANOVA test, with LSD means comparison.

·         I had reservations regarding this choice. Is LSD the most suitable method for comparing four-year seasonal averages? The Fisher's LSD test essentially conducts individual t-tests for all pairwise comparisons between group means, but it does not incorporate any adjustment for the error rate associated with multiple comparisons.

3.       Data management and statistical evaluation. Considering the objectives and design of the study, it is necessary to provide a detailed description of the statistical analysis methods and indicators used.

4.       Results. Discussion.  From my perspective, the analysis and discussion of the results in the article are both thorough and detailed, effectively aligning with the study's intended objectives.

5.       Conclusions The authors discovered that a significant proportion of raw cow milk samples collected in Gorgan, Golestan province, located in the northeast of Iran and the southeast of the Caspian Sea, contained AFM1, with a prevalence rate of 77.2%. Among these samples, 14.4% and 22.7% were found to exceed the regulatory limits set by INSO and the EU, respectively.

They also offer recommendations to address the issue of AFM1 contamination in dairy products. These recommendations include the production of high-quality animal feed and regular inspections of milk and milk products.

·         However, while this knowledge is essential for the region, it may not be groundbreaking on an international scale. Therefore, I would suggest that the authors consider what aspects of their research could be relevant and valuable to farmers and scientists in other countries.

Specific comments

1.       In my view, there is room for expansion in the statistical analysis of the data, particularly with regards to comparing sample proportions across different seasons. This could be achieved through the application of non-parametric analysis methods. Expanding the statistical analysis in this manner would enhance the comprehensiveness of the study and contribute to a more robust evaluation of the impact of seasonal variations on AFM1 prevalence in milk samples.

2.       The bibliography must be carefully reviewed and prepared according to the requirements of the journal.

Conclusion. The authors have taken a comprehensive approach to presenting their findings and have provided a clear interpretation of the data in relation to the research goals. Furthermore, their discussion delves into the implications of these findings, shedding light on the potential risks associated with AFM1 contamination in raw cow milk in the specific region of Gorgan, Golestan province, Iran. They have aptly highlighted the instances where regulatory limits were exceeded, emphasizing the importance of monitoring and quality control in the dairy industry.

However, one aspect that warrants consideration is the broader relevance of this research on a global scale. While the study's findings are undoubtedly significant for the local context, it's worth contemplating how these results and insights might contribute to the knowledge base of farmers and scientists in other regions and countries facing similar challenges.

Sincerely, reviewer.

Author Response

Reviewers 3

Dear Esteemed Reviewer,

Greetings,

I wish to express my heartfelt gratitude for dedicating your valuable time to review our manuscript titled "Seasonal study of Aflatoxin M1 contamination in cow milk of retail dairy market in Gorgan, Iran". Your insightful feedback and profound insights have played a pivotal role in enhancing the quality of our research. We deeply appreciate your recognition of our work. Your feedback has been constructive, and we will undertake the necessary revisions to ensure a more comprehensive documentation of our methods, thus promoting transparency and reproducibility.

Once again, thank you for investing your time, effort, and expertise in the review of our manuscript.

Sincerely Yours,

AS Mozaffari Nejad,

On behalf of other Authors

Comments

Brief summary. Dairy products, particularly milk, are essential components of nutrition. However, it's crucial to be aware that milk can potentially pose health risks due to the presence of a harmful substance known as Aflatoxin M1 (AFM1). This study aimed to evaluate the AFM1 levels in cow's milk available in the retail dairy market of Gorgan, Iran. The assessment was conducted in accordance with the guidelines established by the European Union, the U.S. Food and Drug Administration, and the Iranian National Standards Organization (INSO). The accomplishment of these objectives holds practical relevance, particularly concerning food safety in the mentioned region and its implications for human health.

General concept comments

  1. The introduction provides an overview of literature sources related to the examined topic (22 articles), outlines the article's objectives, with an emphasis on highlighting the practical significance of the conducted research.

However, I noticed that there is a lack of a clear hypothesis formulation and the authors' perspective on how their article could be relevant in theoretical terms to the international scientific community?

  • Thank you so much for considering this subject. We appreciate your feedback regarding the lack of a clear hypothesis formulation and our perspective on the relevance of our article to the international scientific community. We acknowledge that although we mentioned some of these points in the conclusion section. We have taken your valuable and excellent feedback into account and made the necessary changes to the conclusion section. We have revised and restructured the content to provide a more concise and compelling statement of our hypothesis and to highlight the significance of our research in theoretical terms to the international scientific community. Once again, we genuinely appreciate your input, and we are grateful for the opportunity to improve our manuscript based on your suggestions.
  1. Materials and Methods. The "Materials and Methods" section encompassed several subsections: Sampling, AFM1 analysis by ELISA, Sample preparation, ELISA test procedure, and Statistical analysis. The sample consisted of 180 raw cow milk samples. Statistical analysis employed the ANOVA test, with LSD means comparison.

I had reservations regarding this choice. Is LSD the most suitable method for comparing four-year seasonal averages? The Fisher's LSD test essentially conducts individual t-tests for all pairwise comparisons between group means, but it does not incorporate any adjustment for the error rate associated with multiple comparisons.

  • We are sincerely grateful for your detailed feedback and attention regarding the statistical aspect of our results. However, after submitting the article to the journal, we discovered an error and immediately took steps to rectify it. We must acknowledge that at the time of submission, some of our team members were on summer vacation, making it challenging to access the necessary resources. Additionally, we were under pressure to meet the submission deadline, leaving us with limited time to thoroughly review the article. We apologize for this oversight, which stemmed from our previous workload. We greatly appreciate your diligence and careful review. Following the esteemed referee's opinion, we conducted a thorough re-evaluation of the results. We respected your feedback and have duly rectified the writing mistake as per your suggestion. We hope you find our explanation reasonable. Once again, we extend our heartfelt gratitude for your attention to detail.
  1. Data management and statistical evaluation. Considering the objectives and design of the study, it is necessary to provide a detailed description of the statistical analysis methods and indicators used.
  • We appreciate the reviewer's feedback regarding the need for a detailed description of the statistical analysis methods and indicators used in the study. We apologize for the lack of clarity in the statistical section and appreciate the opportunity to provide additional information. For the statistical analysis, we utilized SPSS software version 19 (ISPSS Chicago, IL, USA). The mean level of AFM1 in raw milk samples collected during different seasons (Winter, Spring, Summer & Autumn) were compared using an ANOVA test. Furthermore, we compared the mean concentration of AFM1 in the samples with the permitted amounts of this mycotoxin according to the INSO and EU regulations, which are 100 ng/L and 50 ng/L, respectively. To determine statistical significance, we used a significance level (α) of P < 0.05. We apologize for the lack of detail initially provided and have now added this information to the statistical section of the manuscript to ensure transparency and a better understanding of the analysis methods employed. We sincerely thank the reviewer for highlighting this issue and allowing us the opportunity to improve the clarity and comprehensibility of our study.
  1. Results. Discussion. From my perspective, the analysis and discussion of the results in the article are both thorough and detailed, effectively aligning with the study's intended objectives.
  • Thank you for your attention and feedback.
  1. Conclusions The authors discovered that a significant proportion of raw cow milk samples collected in Gorgan, Golestan province, located in the northeast of Iran and the southeast of the Caspian Sea, contained AFM1, with a prevalence rate of 77.2%. Among these samples, 14.4% and 22.7% were found to exceed the regulatory limits set by INSO and the EU, respectively.

They also offer recommendations to address the issue of AFM1 contamination in dairy products. These recommendations include the production of high-quality animal feed and regular inspections of milk and milk products.

However, while this knowledge is essential for the region, it may not be groundbreaking on an international scale. Therefore, I would suggest that the authors consider what aspects of their research could be relevant and valuable to farmers and scientists in other countries.

  • Thank you so much for considering this subject. We highly appreciate your feedback regarding the lack of consideration for the aspects of our research that could be relevant and valuable to farmers and scientists in other countries. While we did mention some of these points in the current section, we understand the need for a more comprehensive and explicit discussion. We have carefully considered your valuable and excellent feedback and have made the necessary changes to the conclusion section. We have revised and restructured the content to provide a more concise and compelling statement of our hypothesis, as well as to highlight the significance of our research in theoretical terms to the international scientific community. Once again, we genuinely appreciate your input and are grateful for the opportunity to improve our manuscript based on your insightful suggestions.

Specific comments

  1. In my view, there is room for expansion in the statistical analysis of the data, particularly with regards to comparing sample proportions across different seasons. This could be achieved through the application of non-parametric analysis methods. Expanding the statistical analysis in this manner would enhance the comprehensiveness of the study and contribute to a more robust evaluation of the impact of seasonal variations on AFM1 prevalence in milk samples.
  • Thank you for your attention and feedback.
  1. The bibliography must be carefully reviewed and prepared according to the requirements of the journal.
  • Checked. 

Conclusion: The authors have taken a comprehensive approach to presenting their findings and have provided a clear interpretation of the data in relation to the research goals. Furthermore, their discussion delves into the implications of these findings, shedding light on the potential risks associated with AFM1 contamination in raw cow milk in the specific region of Gorgan, Golestan province, Iran. They have aptly highlighted the instances where regulatory limits were exceeded, emphasizing the importance of monitoring and quality control in the dairy industry.

However, one aspect that warrants consideration is the broader relevance of this research on a global scale. While the study's findings are undoubtedly significant for the local context, it's worth contemplating how these results and insights might contribute to the knowledge base of farmers and scientists in other regions and countries facing similar challenges.

  • Revised. 

Round 2

Reviewer 1 Report

The authors have explained all issues raised and now I think the manuscript is suitable for publication.